Research on channel estimation based on joint perception and deep enhancement learning in complex communication scenarios

Liu Xin 1 2 18165473619@189.cn
Zhao Shanghong 2
Liang Yanxia 3
http://orcid.org/0000-0001-9986-5052 Karim Shahid 1
1 School of Information Engineering, Xi’an Eurasia University , Xi’an , China
2 School of Information and Navigation, Air Force Engineering University , Xi’an , China
3 School of Communications & Information Engineering, Xi’an University of Posts & Telecommunications , Xi’an , China
Kong Xiangjie
Electronic publication date: 2025 May 8
Publication date: 2025
Volume: 11
Electronic Location ID: e2852
Received 2024 Dec 12; Accepted 2025 Apr 2
Copyright: © 2025 Liu et al.
Copyright year: 2025
Copyright holder: Liu et al.
License: This is an open access article distributed under the terms of the Creative Commons Attribution License, which permits unrestricted use, distribution, reproduction and adaptation in any medium and for any purpose provided that it is properly attributed. For attribution, the original author(s), title, publication source (PeerJ Computer Science) and either DOI or URL of the article must be cited.
License URL: https://creativecommons.org/licenses/by/4.0/

Keywords: Deep learning, IRS, Channel estimation, DDPG

Funding: The authors received no funding for this work.

==============================
In contemporary wireless communication systems, channel estimation and optimization have become increasingly pivotal with the growing number and complexity of devices. Communication systems frequently encounter multiple challenges, such as multipath propagation, signal fading, and interference, which may result in the degradation of communication quality, a reduction in data transmission rates, and even communication interruptions. Therefore, effective estimation and optimization of channels in complex communication environments are of paramount importance to ensure communication quality and enhance system performance. In this article, we address the intelligent, reflective surface (IRS)-assisted channel estimation problem and propose an intelligent channel estimation model based on the fusion of convolutional neural network (CNN) and gated recurrent unit (GRU) row features, utilizing the reinforcement learning Deep Deterministic Policy Gradient (DDPG) strategy for Channel Reconstruction Prediction and Generation Network (CRPG-Net). The framework initially acquires the received signal by converting the guide-frequency symbols at the transmitter into time-domain sequences to be transmitted, and after propagating through the direct channel and the IRS reflection channel, processes the data at the receiver. Subsequently, the spatial and temporal features in the received signal are extracted using the CRPG-Net model, with the adaptive optimization capability of the model enhanced by deep reinforcement learning. The introduction of reinforcement learning enables the model to continuously optimize decisions in dynamic channel environments, improve the robustness of channel estimation, and quickly adjust the IRS reflection parameters when the channel state changes to adapt to complex communication conditions. Experimental results demonstrate that the framework achieves significant channel estimation accuracy and robustness across several public datasets and real test scenarios, with the channel estimation error markedly smaller than that of traditional least squares (LS) and linear minimum mean square error (LMMSE) methods. This method introduces innovative techniques for channel estimation in intelligent communication systems, playing a crucial role in enhancing communication quality and overall system performance.

Introduction

Communication technology has undergone profound development over the past few centuries. From the advent of telegraph and telephone technology to the evolution of radio communication, satellite communication, and the contemporary era of mobile communication and the Internet, each technological leap has significantly transformed human life and social structure. Modern communication technologies, particularly in recent decades, have seen rapid advancements, especially the iterative evolution of mobile communication technologies from 1 to 5G, and the forthcoming 6G, continually enhancing communication speed and data transmission capacity, thus significantly improving user experience. Among these advancements, channel estimation emerges as a pivotal technology in wireless communication systems (Xiao et al., 2024). The precision of channel estimation directly influences the performance of the communication system, including data transmission rate, bit error rate, and signal coverage. Intelligent, reflective surface (IRS) is an innovative technology in the communication field, capable of dynamically adjusting the propagation path of electromagnetic waves by incorporating controllable reflecting units, thereby enhancing signal quality and coverage. The introduction of the IRS not only boosts the overall performance of the communication system but also offers a cost-effective and energy-efficient channel optimization solution (Hu et al., 2024). However, accurate channel estimation and real-time reflection unit configuration are crucial to fully exploit the potential of the IRS. Traditional methods such as least squares (LS) and linear minimum mean square error (LMMSE), although computationally simple and easy to implement, have significant limitations when facing complex channel environments. For example, in situations where multipath effects are significant, channel states change rapidly, or there is strong interference, these methods often struggle to accurately capture channel characteristics, leading to a decrease in estimation accuracy. In addition, due to its reliance on a fixed linear model, it is difficult to effectively adapt to the dynamic configuration requirements of the IRS and cannot achieve adaptive optimization in rapidly changing communication environments, thereby limiting the performance improvement of the system. Therefore, achieving high-precision channel estimation in complex communication environments has become a focal point and challenge in current research.

The advent of IRS technology offers novel solutions for addressing channel issues in communication systems. By incorporating adjustable reflective units in the signal propagation path, IRS can effectively manipulate the signal’s trajectory, enhancing its strength and stability. This technique is particularly crucial in non-line-of-sight (NLOS) propagation environments and scenarios plagued by severe multipath effects. It not only significantly improves channel capacity and coverage but also reduces the system’s energy consumption (Ge et al., 2023). However, despite the considerable theoretical advantages of the IRS, its practical application encounters numerous challenges, particularly in channel estimation. Traditional methods, such as LS and LMMSE, although straightforward and easy to implement, lack sufficient accuracy in complex channel environments and fail to meet the dynamic configuration requirements of the IRS. To address these limitations, researchers explore the integration of machine learning and deep learning techniques into channel estimation. Machine learning methods, such as support vector machines (SVM) and random forests (RF), can enhance channel estimation accuracy by leveraging extensive historical data. Nonetheless, these methods still grapple with challenges like feature selection and high data dimensionality (Soltani et al., 2024). Deep learning methods, including convolutional neural networks (CNN), recurrent neural networks (RNN), and Transformers, have emerged as research focal points for channel estimation due to their robust feature extraction and nonlinear fitting capabilities. These methods can automatically extract features in complex channel environments through end-to-end learning, significantly improving estimation accuracy (Jiang, Cheng & Yu, 2021).

The integration of IRS and deep learning methods can not only markedly enhance the accuracy and efficiency of channel estimation but also facilitate adaptive optimization in complex communication environments. IRS can substantially improve signal quality and coverage in multipath propagation and NLOS settings by dynamically adjusting the propagation paths of electromagnetic waves. Meanwhile, deep learning techniques can effectively address nonlinearities and dynamic fluctuations in intricate channel environments through automatic feature extraction and end-to-end learning. The synergy between IRS and deep learning introduces innovative solutions for the advancement of future wireless communication technologies. In this context, this article investigates channel estimation based on the amalgamation of intelligent, reflective surfaces and deep reinforcement learning methods in complex communication scenarios, with the following contributions:

1. This study successfully extracts the spatial and temporal features of channel state information (CSI) in complex communication environments by constructing a Channel Reconstruction Prediction and Generation Network (CRPG-Net) based on CNN and gated recurrent unit (GRU). The CNN effectively extracts local spatial features in the received signals, while the GRU captures the dynamic features of the channel over time, addressing the challenging problem of channel estimation in multipath propagation and dynamic channels. The multi-level feature extraction capability of CRPG-Net significantly enhances the accuracy of channel estimation.

2. Utilizing the features extracted from CRPG-Net as state inputs to the deep reinforcement learning algorithm, namely Deep Deterministic Policy Gradient (DDPG) algorithm, this study designs an effective reward function to optimize the configuration of reflective units of the IRS. The DDPG model continuously learns and adjusts the IRS configuration strategy in a dynamic environment, adaptively optimizing the channel propagation path and effectively improving signal quality and coverage. This combination not only enhances real-time channel estimation but also significantly boosts the overall performance of the communication system.

3. The proposed CRPG-Net method is validated across several public datasets and real test scenarios. Experimental results demonstrate that CRPG-Net significantly surpasses traditional LS and LMMSE methods in channel estimation, with markedly improved robustness and adaptability in complex environments.

The remainder of this article is organized as follows: “Related Works” introduces related works on IRS and channel information estimation. In “Methodology”, the CRPG-Net framework is established. “Experiment Result and Analysis” presents the experiment and result analysis. Discussion is provided in “Discussion”, followed by the conclusion.

Related works

Joint sensing and intelligent reflective surface channel estimation

Joint sensing (JS) typically pertains to the field of wireless communication and signal processing, where multiple sensors or devices collaborate to sense and acquire information about the environment or channel state. This approach facilitates more accurate and comprehensive environment sensing results through integrated processing and analysis of information (Chen et al., 2020). The U.S. Defense Advanced Research Projects Agency explores spectrum-sharing methods for joint perception communication systems, aiming to share radar operating frequency bands for communication system use while ensuring the normal operation of military radars, thereby enhancing the available bandwidth and performance of military and commercial communications. There is spectrum sharing between radar and communication systems in many signal bands, with most spectrum resources below 10 GHz predominantly used by radar systems. New technologies for wireless communication systems, such as 5G, long term evolution (LTE), and WiFi, are sharing the spectrum in this band (Liu et al., 2020). Currently, applying artificial noise in the transmitted signal and designing the corresponding beam assignment matrix is a critical tool for physical layer security design. Artificial noise is placed in the zero space of the legitimate user’s channel to interfere with and disrupt the eavesdropper’s channel without affecting the user’s received signal signal-to-noise ratio (SNR) (Alghamdi et al., 2020). Chu et al. (2020) established a single-objective, single-antenna joint sensing communication model, constructing an optimization problem with the objectives of maximizing the minimum secrecy rate of the system, maximizing the return signal SNR, and minimizing the transmit power. They jointly designed a beamforming matrix for the transmit signal and the artificial noise, utilizing the first-order Taylor expansion to handle the non-convex secrecy rate to obtain the suboptimal solution to the problem (Chu et al., 2020). It is evident that joint perceptual techniques are highly complementary to overall communication technology and can effectively improve signal quality. Furthermore, the wireless channel environment is considered a randomly varying entity that degrades received signal quality due to uncontrolled reflections, refractions, and incidental interference (Renzo et al., 2019). To mitigate these negative effects, reconfigurable intelligent surface (RIS), or intelligent reflective surface (IRS) technology, offers a promising innovative solution. Mishra & Johansson (2019) proposed a simple on/off two-state channel estimation protocol, which estimates the cascaded channel corresponding to a RIS reflection element one at a time by sequentially turning on each reflection element. This scheme estimates the cascaded channel with a pilot overhead of M. Jensen & De Carvalho (2020) proposed an LS estimator for the cascaded channel, designing the RIS reflection coefficients phase-shift matrix based on discrete fourier transform (DFT).

Channel estimation based on deep learning methods

Deep learning, as an advanced machine learning technique, can automatically extract complex spatio-temporal features and handle nonlinear relationships in the channel, significantly improving the estimation accuracy of CSI. With large-scale training data, deep learning models exhibit strong noise robustness and maintain highly accurate channel estimation even in complex and variable communication environments. Elbir et al. (2020) were the first to apply a CNN estimator to a RIS-assisted downlink millimeter-wave channel. The estimation is implemented in two stages, somewhat similar to the previously mentioned on/off-based strategy, by assuming that each user has access to the CNN to estimate its channel. A dual CNN is designed to estimate both direct and cascaded channels. Kundu & McKay (2021) models the RIS channel estimation problem as a least-squares solution denoising problem. Mao, Liu & Peng (2022) utilized deep residual networks in RIS-assisted wave channels to improve the channel estimates generated by orthogonal matching tracking. Ye, Zhang & Wang (2022) explored the use of generative adversarial networks (GANs) for RIS-assisted large-scale multiple-input multiple-output (MIMO) estimation in millimetre waves. They used GANs and residual dense networks (RDN) to enhance the performance of channel estimation based on compressed sensing. Gao et al. (2020) employed an unsupervised learning approach to reduce the training overhead associated with supervised learning. They utilized a deep neural network (DNN) to address the problem of passive beamforming design for IRS. Additionally, Huang, Mo & Yuen (2020) proposed using deep reinforcement learning (DRL) for beamforming optimization design, aiming to maximize the total multiuser transmission rate and employing the DDPG algorithm to solve the joint optimization problem for transmitter and reflector surfaces. Through the above research, it can be seen that deep learning has shown strong potential in assisting channel estimation and optimization in IRS. Compared to traditional methods, deep learning can automatically extract complex spatiotemporal features, capture nonlinear relationships in channels, and maintain high estimation accuracy in noisy environments. Existing research indicates that CNN, GAN, deep residual networks and other models have been successfully applied to RIS-assisted millimetre wave channel estimation, while DRL has further promoted IRS beamforming optimization. Overall, deep learning provides intelligent means for IRS channel modelling, estimation, and optimization, which is expected to improve the reliability and efficiency of future wireless communication systems.

Through the above research, it is evident that there is a lack of studies related to orthogonal frequency-division multiplexing (OFDM) technology in RIS and IRS-assisted communication systems compared to millimetre wave technologies. However, as a widely used communication technology in wireless communication, exploring the feasibility of combining OFDM with deep learning technology in IRS-assisted scenarios is crucial for realizing intelligent wireless communication systems in the future. Therefore, optimizing channel estimation using deep learning methods and integrating them with reinforcement learning methods, which are extensively employed in current dynamic environment optimization processes, is an inevitable trend in complex communication scenarios.

Methodology

CNN and GRU

One-dimensional convolutional neural network (1D-CNN) is suitable for processing one-dimensional data. It is widely used in time series analysis, signal processing, and natural language processing. In channel estimation, 1D-CNN performs feature extraction on the input signal through convolutional operations to capture the local features and patterns of the signal. The basic building blocks of 1D-CNN include convolutional layers, activation functions, pooling layers, and fully connected layers, with feature extraction primarily accomplished by the convolutional and pooling layers (Kattenborn et al., 2021).

The convolutional operation is the core of 1D-CNN, extracting local features by sliding the convolutional kernel over the input signal and calculating the weighted sum of the local region, as shown in Eq. (1):

(1) y[i]=(x∗w)[i]+b=∑j=0k−1x[i+j]⋅w[j]+b

where y[i] is the convolution result, x is the input, w is the kernel, b is bias and the convolved result is nonlinearly transformed by the activation function. The activation function used is the rectified linear unit (ReLu) function and pooling is done after this operation. Local feature extraction is performed on the input signal through the convolutional layer, where the convolution operation captures the local patterns and features in the signal. The convolution result is nonlinearly transformed through an activation function, enabling the model to learn more complex features. The pooling layer reduces the size of the feature map, preserving the essential features and reducing computational complexity.

After completing the feature extraction using the CNN approach, we employ GRU to capture signal temporal features further. GRU addresses the problem of gradient vanishing and gradient explosion in standard recurrent neural networks (RNNs) when handling long-time dependencies by introducing a gating mechanism. GRU has a simpler structure and fewer parameters than long short-term memory networks (LSTMs), yet it demonstrates similar performance in many tasks. The basic components of GRU consist of two gates: the reset gate and the the update gate (Cahuantzi, Chen & Güttel, 2023). The computation process of the reset gate is shown in Eq. (2):

(2) rt=σ(Wr⋅[ht−1,xt])

where rt is the output, Wr is the weights, ht−1 is the hidden state, xt is the current input and σ is the sigmoid activation function. The update gate is calculated as shown in Eq. (3):

(3) zt=σ(Wz⋅[ht−1,xt])

where zt is the output and Wz is the weight. Based on this according to the reset gate and update gate to complete the update of the candidate hidden state and the current hidden state, both are calculated as shown in Eqs. (4) and (5):

(4) h~t=tanh(Wh⋅[rt⊙ht−1,xt])

(5) ht=(1−zt)⊙ht−1+zt⊙h~t

where: h~t is the candidate hidden state. Wh is the weight matrix. rt⊙ht−1 denotes the element-by-element product of the output of the reset gate and the hidden state of the previous moment. tanh is the hyperbolic tangent activation function. (1−zt) controls the degree of retention of the hidden state at the previous moment. Through the use of GRU, the temporal correlation of channel state information can be effectively captured to improve the accuracy of channel estimation, which provides an efficient solution for channel estimation in complex communication scenarios.

Policy-based deep reinforcement learning method DDPG

DDPG is a policy gradient-based method designed for problems in continuous action spaces. It combines the advantages of Deep Q-Network (DQN) and policy gradient methods, achieving efficient policy learning by introducing the Actor-Critic architecture and the experience replay mechanism. The main steps of DDPG include initializing the network, acquiring data, updating the Critic network, updating the Actor network, and soft updating of the target network (Xu et al., 2020). Firstly in initializing the network, it is necessary to initialize the Actor network μ(s|θμ) and the Critic network Q(s,a|θQ) and its corresponding target networks μ′(s|θμ′) and Q′(s,a|θQ′) and copy the parameters θμ and θQ to the target network. Execute the current policy in the environment, exploring the state space via ϵ—greedy policy or noise. Store each transfer sample (st,at,rt,st+1) into the experience playback buffer. Next, in performing the update of the Critic network, compute the target Q-value as shown in Eq. (6):

(6) yi=ri+γQ′(si+1,μ′(si+1|θμ′)|θQ′).

Then the minimization of network losses is being performed:

(7) L=1N∑i(yi−Q(si,ai|θQ))2.

The loss function is minimized using gradient descent algorithm as shown in Eq. (8):

(8) ∇θμJ≈1N∑i∇aQ(s,a|θQ)|s=si,a=μ(si)∇θμμ(s|θμ)|s=si.

ri is the reward, γ is the discount factor, and Q′ and μ′ are the target Critic network and target Actor network respectively. So far the update of the target soft network is realized as shown in Eqs. (9) and (10):

(9) θQ′←τθQ+(1−τ)θQ′

(10) θμ′←τθμ+(1−τ)θμ′

where τ is a constant less than 1, which usually takes a small value for smoothly updating the parameters of the target network. For the above update process involved in DDPG, it can be represented by Fig. 1.

Figure 1 The framework for the DDPG.

The establishment for the channel estimation model CRPG-Net

After introducing the basic framework used for the model, we conducted an analysis of the channel variation estimation method based on intelligent reflective surfaces. We constructed the CRPG-Net, which leverages CNN and GRU for feature extraction, and employed a reinforcement learning-based method for dynamic performance enhancement of the model. On this basis, this article completes the channel estimation of IRS-based signals, and the overall flow of the process is illustrated in Fig. 2:

Figure 2 The framework for the CRPG-Net.

As shown in Fig. 2, the guide frequency symbols at the transmitting end are first transformed into the time-domain sequence x(n) to be transmitted by inverse fast Fourier transform (FFT) and data processing. The time-domain sequence x(n) propagates through the direct channel and IRS reflective channel, and then the receiver symbol Y(k) is obtained through data preprocessing and FFT at the receiver. Subsequently, Y(k) is fed into the channel estimation module along with the guide signal X(k). Conventional channel estimation methods use a linear or quasi-linear model in this step. However, in this study, a neural network model, specifically the proposed CRPG-Net, is used instead of the conventional channel estimation module. The CRPG-Net neural network transforms the channel estimation problem into a multivariate regression problem by fitting the mapping relationship between the guide frequency sequence and the received signal through its powerful feature learning capability, thereby achieving rapid optimization and analysis of the effects of the intelligent reflective surface.

Experiment result and analysis

Experiment setup and dataset

In this study, we used the COST 2100 Channel Model as the channel data source, which can simulate wireless channel characteristics in different environments, such as indoor scenes and urban microcells, and provide channel state information (CSI) for multi-user and multi antenna systems (Flordelis et al., 2019). Extract training and testing sets within each category in a 7:3 ratio to ensure consistent distribution of training and testing data in different channel environments and avoid affecting the model’s generalization ability due to uneven partitioning. In addition, to ensure that the model can fully learn channel characteristics, we randomly shuffle the original data before data partitioning, while ensuring that the pilot signal and corresponding channel state information always appear in pairs to avoid information leakage or mismatch. To improve data quality, we normalize CSI data to reduce the impact of numerical fluctuations on the model, and use dimensionality reduction methods to compress data dimensions and reduce computational complexity. In addition, by adding noise disturbances and constructing temporal windows, the adaptability of the model to channel changes is enhanced, ensuring that CRPG-Net can efficiently learn channel characteristics and improve estimation accuracy.

For model evaluation, we employed the normalized mean square error (NMSE) as the performance metric. Taking a typical direct channel Cd as an example, whose calculation process is shown in Eq. (11):

(11) NMSECd=1J∑j=1JCdj−C^d(j)Cd(j).

J denotes the number of Monte Carlo simulations, and C^d(j) represents the channel estimate obtained from each simulation. In the channel estimation process, we used several methods for comparison, including the traditional channel prediction methods LS (Demir, Bjornson & Sanguinetti, 2022), LMMSE (Wu, 2021), and the CNN-GRU method without DDPG reinforcement. Additionally, considering the involvement of deep learning models, we constructed the experimental environment, with the specific parameters shown in Table 1.

Table 1 The experiment environment information.

Environment	Information	
CPU	I7-14700F	
GPUs	RTX 4060Ti	
Language	Python 3.5	
Framework	Pytorch	

Method comparison and result analysis

After completing the model construction and dataset confirmation, and depersonalizing the data inputs and outputs, we performed a comparative analysis of the models. The results of the channel prediction under the common dataset are shown in Fig. 3.

Figure 3 Channel estimation results under COST 2100 channel model dataset.

We searched for signals with different signal-to-noise ratios (SNR) and the corresponding channel results in the dataset, analyzing signals with varying SNRs. As shown in Fig. 3, with increasing SNRs under the exponential coordinate system, the overall channel prediction performances of both the traditional methods and the method proposed improve. The proposed CRPG-Net achieves better channel performance estimation due to the inclusion of the reinforcement learning module, surpassing traditional channel estimation methods such as LMMSE.

To compare the overall performance under different SNRs more intuitively, we calculated the NMSE mean values of different models. The results, shown in the bar charts, indicate that the proposed CRPG-Net method exhibits a more stable channel estimation effect, with overall performance significantly better than that of the traditional LS method and the CNN-GRU method without the reinforcement learning addition.

The real test for the IRS optimization for the channel estimation

After completing data testing under the public dataset, we conducted practical testing to evaluate not only the performance of the model on channel prediction but also the resultant performance of the intelligent reflective surface under different conditions. The overall structure built is depicted in the overall flow in Fig. 2, where an access point equipped with N antennas is set as the transmitter, a single-antenna user serves as the receiver for communication, and a 2D smart reflective surface is placed between the user and the transmitting access point (AP) to act as an unattached relay node. Due to severe path loss, only the signal reflected once by the smart reflective surface is considered.

The channel form is consistent with the overall signal estimation process described in “The Establishment for the Channel Estimation Model CRPG-Net”, where the signal undergoes an N-point FFT into the guided symbols to be transmitted, then is transmitted and processed to form the final signal and the corresponding channel. In this propagation model, we need to estimate the direct channel H and the cascaded reflected channel G during downlink transmission (Guo & Lau, 2022).

Under the simulation experiments performed based on the above information, we obtained the direct channel estimation and cascaded reflection estimation results, as shown in Fig. 4.

Figure 4 Plot of direct signal and cascaded reflection channel estimation results.

The performance of CRPG-Net, the traditional channel estimation method, and the CNN-GRU method with reinforcement learning for estimating direct and cascaded reflective channels under different signal-to-noise ratios is presented in Fig. 4 in the real simulation tests. From the results in the figure, it is evident that CRPG-Net significantly outperforms the traditional LS channel estimation methods and also demonstrates a performance advantage over the high-complexity LMMSE method. Additionally, CRPG-Net performance gradually saturates at high signal-to-noise ratios, indicating that it cannot achieve estimation performance with errors infinitely close to zero. To observe the model effect more intuitively, we calculated and analyzed the mean NMSE values under the two estimations, the results of which are shown in Fig. 5.

Figure 5 Plot of mean values of direct signal and cascaded reflection channel estimation results.

In Fig. 5, it is evident that both in the process of direct channel estimation and cascaded channel estimation, the CRPG-Net method used achieves superior estimation results. Its overall average error remains below one. Additionally, the robustness of the model is analyzed to some extent. In the intelligent reflecting surface (IRS) assisted communication system, the number of reflection elements M represents the number of independently adjustable reflection units on the IRS. Each reflection unit can adjust the phase and amplitude of the incident signal to optimize the signal propagation path and improve communication quality. In theory, increasing M can enhance the IRS’s ability to regulate the wireless environment, enabling the receiving end to obtain better signal quality. However, in practical applications, different M settings may lead to mismatches in reflection coefficients, where the control accuracy and consistency of each IRS reflection unit may be limited by hardware or external environmental interference, thereby affecting the accuracy of channel estimation and system performance. Therefore, in this article, we studied the performance of reflection cascade channel estimation under different M settings, analyzed the channel estimation error changes of IRS under different reflection unit sizes through experiments, and verified the adaptability and robustness of the proposed method under different conditions. This study helps to understand the scale effect of IRS and provides guidance for the selection of the number of IRS units in actual deployment. The results under different M are shown in Fig. 6.

Figure 6 Comparison of cascaded reflection channel estimation performance under different M.

In Fig. 6, we present a comparison of the channel estimation performance of the reflection cascade with different M settings. It can be observed that the channel estimation performance curves of CRPG-Net maintain a similar trend, and the gap between them remains minimal even when the number of reflection elements M of the system reflection surface does not match the number of reflection elements used for the offline training of the model. These results demonstrate that the performance of the proposed CRPG-Net is stable and exhibits good robustness under different IRS settings.

To further explore the performance enhancement brought by smart reflective surfaces to the system, this chapter compares the system achievable rates obtained by different algorithms after beamforming co-optimization design under various transmitter maximum transmission power P constraints. These results are also contrasted with scenarios where smart reflective surfaces are not used as auxiliary communication devices. The results are illustrated in Fig. 7.

Figure 7 Comparison of system achievable rate with different transmission power.

In scenarios where the transmitter transmission power is very low, the system’s achievable rate is limited, regardless of the presence of IRS. As the power is gradually increased, the system’s achievable rate also increases. Furthermore, when comparing the method without reinforcement learning, it is evident that the CRPG-Net used in this article yields better results, achieving a higher achievable rate. Therefore, practical verification demonstrates that using intelligent reflective surfaces for signal enhancement, combined with the introduction of advanced deep learning algorithms, is crucial for accurate channel estimation and overall signal performance improvement.

Discussion

Channel estimation is a crucial component in wireless communication systems, essential for obtaining system channel information and exploring the impact of intelligent reflective surfaces as a novel tool for enhancing communication performance. In this study, we propose CRPG-Net, which integrates CNN and GRU for feature extraction and combines them with a reinforcement learning approach for dynamic performance enhancement in intelligent channel estimation. Experimental results from public datasets and real-world tests demonstrate that CRPG-Net significantly outperforms traditional methods such as LS and LMMSE. Traditional LS and LMMSE methods are predominantly based on linear or quasi-linear models, making them less effective at capturing the nonlinear and dynamic changes of channel characteristics in complex environments. CRPG-Net leverages the feature extraction capabilities of CNN and GRU. CNN, with its powerful local feature extraction capability, can efficiently capture the spatial features of the received signal, while GRU further models temporal dependencies, enhancing its ability to model dynamic channel changes and maintaining stable estimation performance even in the face of rapidly changing channels. In addition, reinforcement learning (DDPG) optimizes strategies through continuous interaction, allowing the IRS reflection parameters to adaptively adjust according to the actual channel environment, thereby further improving the robustness of channel estimation and overall system performance. The DDPG algorithm further enhances the model by dynamically tuning the parameters to optimize channel estimation performance. This adaptive learning mechanism allows CRPG-Net to maintain high channel estimation accuracy across different environments. In scenarios involving complex multipath and NLOS propagation, the performance of LS and LMMSE methods degrades significantly. CRPG-Net, however, effectively fits the complex relationship between the guide frequency sequence and the received signal through the neural network’s nonlinear mapping capabilities, transforming the channel estimation problem into a multivariate regression problem. This approach enhances the accuracy and robustness of the estimation.

In the practical tests conducted, we not only evaluated the traditional model’s performance but also examined the performance of different elements of IRS and its various positions. IRS, as an emerging technology, dynamically adjusts the propagation path of electromagnetic waves to enhance signal quality and coverage by introducing controllable reflective units in the channel propagation path. However, the efficient application of IRS relies on accurate channel estimation and real-time reflection unit configuration. The simulation experimental results demonstrate the feature extraction and nonlinear mapping capabilities of CRPG-Net, which can accurately estimate CSI in complex environments, providing a reliable basis for IRS configuration. Additionally, the reflection unit configuration of IRS is dynamically adjusted according to real-time CSI to ensure the optimal selection of the signal propagation path, thereby improving the overall system performance. In complex communication environments, CRPG-Net combined with IRS effectively addresses multipath effects and NLOS propagation issues, enhancing the robustness and reliability of the system. Therefore, the application of IRS, reasonably optimized and combined with advanced deep learning models like CRPG-Net, is of great significance for the future development of communication technology. In dynamic decision-making, delay issues mainly affect the real-time performance of channel estimation and the rapid optimization of IRS. To solve this problem, a temporal modeling mechanism such as LSTM or Transformer can be introduced in CRPG-Net to learn temporal dependencies and predict future channel states. In addition, distillation learning or lightweight models can be used to accelerate inference and reduce computational latency. At the same time, reinforcement learning is introduced to optimize the IRS reflection coefficient, enabling it to make better decisions even when some channel information is missing or lagging, thereby improving the robustness and adaptability of the system in the next generation wireless communication system assisted by IRS. This method can effectively improve channel estimation accuracy, optimize IRS reflection parameters, enhance the system’s adaptability to complex dynamic environments, thereby improving communication quality and data transmission stability. In scenarios such as the Internet of Things (IoT), smart cities, and millimeter wave communication, this method can be used to optimize wireless channel resources, improve spectrum utilization, and support efficient and low latency deployment of intelligent communication networks.

Conclusion

The CRPG-Net, based on CNN with GRU feature extraction and DDPG, proposed in this study, offers an effective solution to the channel estimation problem in complex communication environments. By integrating CNN and GRU techniques and incorporating a deep reinforcement learning strategy, we constructed an estimation system capable of processing and analyzing intricate channel characteristics, achieving high-precision channel estimation assisted by IRS. The experimental results validate the model’s superior performance in practical applications, with significantly better channel estimation error and other evaluation metrics compared to traditional methods such as LS and LMMSE. In tests on public datasets and real scenarios, CRPG-Net demonstrated a substantial improvement in channel estimation accuracy, with NMSE values below one at different signal-to-noise ratios, proving its potential for application in communication systems. This research not only advances wireless communication technology but also provides new technical means and strategic support for the design and optimization of communication systems.

In our future research, we will further explore the application scope of the model by incorporating more diverse communication scenario data and channel environments to enhance its utility and generalization value in the field of wireless communication. Additionally, we plan to introduce advanced AI techniques, such as meta-learning and adaptive algorithms, to improve the model’s adaptability and generalization capabilities. By continuously refining and optimizing the model, we aim to provide more reliable and effective technical support for channel estimation and optimization in wireless communication systems, thereby promoting sustainable development in communication technology.

Supplemental Information

Supplemental Information 1 Code.

Additional Information and Declarations

Competing Interests

The authors declare that they have no competing interests.

Author Contributions

Xin Liu conceived and designed the experiments, analyzed the data, authored or reviewed drafts of the article, and approved the final draft.

Shanghong Zhao conceived and designed the experiments, analyzed the data, prepared figures and/or tables, and approved the final draft.

Yanxia Liang performed the experiments, performed the computation work, prepared figures and/or tables, and approved the final draft.

Shahid Karim performed the experiments, performed the computation work, authored or reviewed drafts of the article, and approved the final draft.

Data Availability

The following information was supplied regarding data availability:

Code is available in the Supplemental Files.

The COST2100 dataset is available at Kaggle: https://www.kaggle.com/datasets/forment/cost2100 (doi: https://doi.org/10.1109/MWC.2012.6393523).

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
