# Peer review of "Research on channel estimation based on joint perception and deep enhancement learning in complex communication scenarios"

_PeerJ Computer Science, doi:10.7717/peerj-cs.2852_

## Round 0.1 · original submission · Major Revisions

Please consider the comments carefully and revise the article accordingly. Then the revised version will be evaluated again.

Reviewer 1 ·

Basic reporting

The author has done a good job of explaining the novelty of the paper and the areas tackled through the abstract and introduction.

The models used such as CNN-1D, GRU, and reinforcement learning were explained with a detailed reasoning behind why they were used in the research.

Experimental design

The research paper is within the aims and scope of the journal.

The authors have diligently defined the research question ie: estimating and optimizing wireless channel communication through deep neural network models and RL.

The framework of the DDPG could be explained further as an algorithm, which will be easy for the readers to follow.

Addressing the latency incurred in the dynamic decision and suggesting how to improve latency or real-time throughput could help address the gap in the implementation. In case the optimization is happening after the signal processing, it needs to be established as to the order of events/ actions that was missing in the paper.

Validity of the findings

Validating the results through the code and dataset shared was difficult as the results were not printed in the notebooks shared nor the dataset was easily readable format from the Kaggle link provided.

After reviewing the paper and analyzing the tables and charts, it's clear that the author has put in considerable effort to compare the results against the benchmark, particularly about SNR and NMSE.

The conclusion effectively summarizes the key points of the paper, and the proposed future work appears to be a logical next step for continuing this research.

·

Basic reporting

1. The background is well-developed but could emphasize the knowledge gap more clearly. Add recent studies from 2023-2024 to enhance the relevance.
2. Follows a logical structure with relevant figures and tables. Figures need more descriptive captions for clarity.

Experimental design

1. Research question is Clearly defined and meaningful. The integration of IRS with deep reinforcement learning is novel.

Validity of the findings

1. Results are robust but they may require additional statistical metrics (e.g., confidence intervals).
2. This study is demonstrated on public datasets, however real-world validation is needed for broader application.

Additional comments

This is a very nice study, I thoroughly enjoyed reading it. I believe this has a great potential to advance the field. Refer below peer review comments for making it reachable to broader audience.

Abstract:

The abstract provides a good summary

1. Include a sentence highlighting the practical significance of integrating CRPG-Net with reinforcement learning for channel estimation.
2. Rephrase this for better readability - "The method presented provides novel technical means for channel estimation in intelligent communication systems and holds great significance for improving communication quality and system performance"


Introduction:
The introduction is comprehensive

1. Can you expand on the limitations of traditional methods (e.g., LS and LMMSE) in complex multipath propagation scenarios.
2. Can you add a comparison between existing deep learning-based models (like CNNs and RNNs) and the proposed CRPG-Net to underscore its advantages.
3. Can you highlight the novelty of combining CNN-GRU features with reinforcement learning using DDPG in the context of IRS-assisted communication.


Methodology
The methodology is detailed

1. Can you specify how the data partitioning (7:3 ratio) was implemented and whether stratification was used.
2. Can you provide more detailed mathematical formulations for the CNN-GRU feature extraction process.

Results
The results section is well-structured

1. Can you also include confidence intervals for the NMSE values shown in the bar charts to enhance statistical robustness.
2. If possible, can you expand the comparison in Figure 3 to include more competing methods (e.g., Transformer-based models or recent hybrid approaches).
3. Can you clearly state the performance metrics used to evaluate robustness under different SNR conditions (Figure 6).


Limitations
The limitations section is briefly touched

1. Can you discuss the generalization of the CRPG-Net framework to scenarios beyond the COST 2100 dataset, especially for real-world use.
2. Can you mention the computational cost of implementing CRPG-Net with reinforcement learning and propose future optimizations to reduce it.


Practical Applications:
While the article discusses applications in IRS-assisted systems, it could benefit from more concrete examples.

1. Provide specific use cases for CRPG-Net, such as its application in urban cellular networks or industrial IoT environments
2. Discuss how the framework could be adapted for 6G communication systems, given its emphasis on dynamic and complex environments.


Conclusion:
The conclusion effectively summarizes the findings

1. Can you suggest future research into using Transformer-based models or meta-learning to enhance the CRPG-Net's adaptability.

References:
1. Are there any additional more recent references to state-of-the-art techniques in channel estimation from 2023 and 2024? If there are, can you please include them.

Reviewer 3 ·

Basic reporting

All comments have been added in detail to the last section.

Experimental design

All comments have been added in detail to the last section.

Validity of the findings

All comments have been added in detail to the last section.

Additional comments

Review Report for PeerJ Computer Science
(Research on channel estimation based on joint perception and deep enhancement learning in complex communication scenarios)

1. Within the scope of the study, a reinforcement learning based deep learning model was proposed to increase the quality of communication in complex communication environments and various channel estimation studies were carried out.

2. In the introduction, intelligent reflective surface technology and the importance of the subject with communication technologies were mentioned at a basic and sufficient level. In addition, the main contributions of the study were stated clearly and in bullet points.

3. In the Related works section, the literature related to the study was discussed in terms of both channel estimation based on deep learning methods and intelligent reflective surface and joint sensing channel estimation. Although the literature on the subject was mentioned in this section, a more in-depth analysis should be made especially in terms of deep learning methods. In this section, it is suggested to add a detailed literature table so that the proposed model can come to the forefront more.

4. In the Methodology section, both gated recurrent unit networks and convolutional neural networks based on deep learning were mentioned at a basic but sufficient level.

5. When the framework and details of the proposed CRPG-Net model are examined and compared with the literature, it is observed that it has a certain level of originality in this study.

6. Examining the types of metrics used in the study and the results obtained accordingly, it is understood that they are both at an acceptable level. In addition, comparing the obtained results with some models in the literature and demonstrating their superiority further increases the quality of the study.

As a result, the study has the potential to present an important deep learning-based model to the literature in terms of channel estimation model. However, attention should be paid to the above sections.

---

## Round 0.2 · accepted · Accept

Many thanks to the authors for their efforts to improve the article. This version satisfied the reviewers successfully. It can be accepted now. Congrats!

·

Basic reporting

This second version looks good.

Experimental design

This second version looks good.

Validity of the findings

This second version looks good.

Additional comments

This second version looks good. I am accepting this.

Reviewer 3 ·

Basic reporting

All comments have been added in detail to the last section.

Experimental design

All comments have been added in detail to the last section.

Validity of the findings

All comments have been added in detail to the last section.

Additional comments

Review Report for PeerJ Computer Science
(Research on channel estimation based on joint perception and deep enhancement learning in complex communication scenarios)

Thanks for the revision. Both the changes made to the paper and the responses to the comments are sufficient. Best regards.